# Effectiveness and safety of an absorbable modified polymer starch powder hemostat versus usual care in gynecology procedures: A prospective, multi-center, and randomized study

Jordi Ponce[1,2], Amparo García-Tejedor [1,2]*, Marc Barahona [1,2], Serena Cappuccio [3], Barbara Costantini[3,4], Camilla Fedele[3], Lola Marti-Cardona[1,2], Giovanni Scambia[3,5]

1 Gynecology Department, University Hospital of Bellvitge (IDIBELL), Hospitalet de Llobregat, Barcelona, Spain, 2 IDIBELL (Instituto de Investigación Biomédica de Bellvitge), Hospitalet de Llobregat, Barcelona, Spain, 3 UOC Ginecologia Oncologica, Dipartimento di Scienze della Salute della Donna, del Bambino e di Sanità Pubblica, Fondazione Policlinico Universitario A. Gemelli, IRCCS, Rome, Italy, 4 UniCAMILLUS, International Medical University, Rome, Italy, 5 Universita' Cattolica del Sacro Cuore, Rome, Italy

* agarciat@bellvitgehospital.cat

## Abstract

### Objective

To evaluate the safety and effectiveness of an absorbable modified polymer starch powder hemostat (AMP-SPH) compared with standard care to control hemostasis when used in adult subjects during open or laparoscopic gynecological procedures for both benign and malignant diseases. Methods: Prospective, multi-center, randomized, and interventional phase IV study conducted on consecutive patients, aged ≥18 years, who underwent an open or laparoscopic gynecological procedure between November 2015 and July 2017 in a third-level Hospital. Study participants were randomly assigned (1:1) to receive either treatment with an AMP-SPH (AMP group) or usual standard care (SC group). The hemostatic treatment administered to the SC group was at the investigator's discretion. The primary effectiveness endpoint of the study was the achievement of hemostasis (yes/no) within 10 minutes. Results: Ninety patients, 44 (48.9%) in the AMP group and 46 (51.1%) in the SC group were included in the analysis. the AMP group, 97.7% (43/44) of patients achieved hemostasis as compared to the 93.5% (43/46) of subjects in the SC group (mean difference: 4.2%; 95%CI: −4.1% to 12.6%; p = .337). The time required to achieve hemostasis was lower in the AMP group (1.91 ± 1.15 minutes) than in the SC group (2.28 ± 2.09 minutes), although not significant (p = .309). A higher proportion of patients in the SC group (17.9%) was observed to require blood products compared to those in the AMP group (4.8%).A total of 29 adverse events (AEs) (24 non-serious and 5 serious AEs) were reported, 12 AEs in the AMP group and 17 in the SC group.

**Data availability statement:** Our study utilizes third-party data that are the property of Artivion, formerly known as CryoLife Europa. As these data are proprietary, they cannot be publicly shared. However, interested researchers may contact Artivion directly to inquire about potential access to the data (https://artivion.com/; 1655 Roberts Blvd., NW; Kennesaw, GA 30144 USA) E-mail address: customerservice.us@artivion.com.

**Funding:** Cryolife, Europe Baxter (Founded Medical writing). The funders had no role in study design, data collection and analysis, decision to publish, or preparation of the manuscript.

**Competing interests:** Barbara Costantini, Jordi Ponce and Giovanni Scambia served as speakers for Baxter and have received speaker honoraria according to fair market value. This does not alter our adherence to PLOS ONE policies on sharing data and materials.

## Conclusions

According to the results of this study, AMP-SPH was not inferior to standard care in the control of bleeding for patients undergoing gynecology procedures and the cessation of bleeding was trending to be faster with the use of AMP-SPH than with standard methods.

## Trial registration

ClinicalTrials.gov NCT02835391.

## Introduction

Hysterectomy is one of the most performed gynecological, surgical procedures worldwide and is second only to cesarean section, with high rates reported in many countries [1–5]. Other surgical procedures are frequently performed in gynecology for both benign and malignant pathologies, such as myomectomy, ovarian cystectomy, endometrial resection or ablation, lymphadenectomy, and others [6,7].

All gynecological surgical procedures carry a bleeding risk, and intraoperative hysterectomy hemorrhage rates of 1% to 2% have been reported [8]. These rates may vary depending on the surgical procedure (0.07%−6.98%) [9], and the presence of underlying medical conditions (i.e., the use of anticoagulant drugs or coagulation disorders).

There are several techniques that surgeons may use to achieve hemostasis when bleeding occurs, including hemoclips or electrocautery. However, in many situations, such as diffuse small vessel bleeding or bleeding near vital structures, these techniques are not sufficient, and hemostatic agents can be used as an adjunctive tool to obtain hemostasis without collateral damage to vital tissue [10–12]. The use of topical hemostatic agents has been shown to not only reduce thermal damage, devascularization, and tissue necrosis, but to also reduce time to hemostasis, and potentially prevent conversion to laparotomy during minimally invasive surgery [10,11].

The aim of this study was to evaluate the safety and effectiveness of an absorbable modified polymer starch powder hemostat (AMP-SPH) compared to usual care when used in adult subjects during open or laparoscopic gynecological procedures in the management of benign or malignant diseases. This study also aimed to assess the clinical outcomes of the AMP-SPH during gynecological procedures.

## Methods

### Study design

This prospective, multi-center, randomized, and interventional phase IV study conducted on patients who underwent an open or laparoscopic gynecological procedure between November 2015 and July 2017.

The study protocol was approved by the Ethic Committee of the Hospital Universitari de Bellvitge and the Ethics Committee of the Policlinico Universitario Agostino Gemelli, and was registered (http://www.clinicaltrials.gov; NCT02835391). This clinical

study was conducted in compliance with the protocol, International Conference on Harmonization – Good Clinical Practice (ICH-GCP) and all applicable regulatory requirements. Participating investigative sites were responsible for complying with applicable regional or national regulations governing the conduct of post market surveillance (follow-up) studies.

Prior written consent was obtained for all patients who were enrolled into the study before any study-specific procedures were performed. To ensure patient confidentiality, any information that could lead to an individual being identified has been encrypted or removed, as appropriate.

### Study participants

This study included consecutive patients, aged ≥18 years; who planned to undergo a gynecological procedure, such as hysterectomy, ovarian cystectomy, myomectomy, endometrial excision or ablation; signed the informed consent; experienced generalized oozing or mild to moderate intraoperative bleeding which was consistent with the requirement for a hemostatic treatment and were willing to comply with the investigators and protocol procedures.

Surgical routes accepted were both open and minimally invasive, including traditional laparoscopic and robotic approaches.

Subjects with a history of pelvic or abdominal radiotherapy within 8 weeks prior inclusion; with a previous ruptured ectopic pregnancy; medical history of abnormal coagulopathy or bleeding; sensitivity to starch or starch derived materials; potential infection at the surgical site; enrolled in another trial at the time of the study; and pregnancy or lactation were excluded.

Additionally, subjects with any major intraoperative bleeding incidences (i.e., American College of Surgeons Advanced Trauma Life Support Class II, III or IV Hemorrhage) (24) were excluded from the study.

### Study groups

Subjects who met eligibility criteria were randomly assigned to receive either treatment with an AMP-SPH (PerClot® Polysaccharide Hemostatic System, Baxter Healthcare Corporation Deerfield, IL 60015 USA) (AMP group) or usual standard care (SC group).

In patients randomized to the AMP group, and accordingly to the manufacturer instructions for use (IFU), a liberal amount of the hemostatic agent was applied directly to the site and source of the bleeding [13].

It was at the Investigator's discretion what treatment they choose for subjects randomized to receive usual care. Hemostatic treatments administered as part of usual care included diathermy or electrocautery, or two commonly used gelatin-thrombin-based flowable hemostatic agents, namely Floseal® (Baxter Healthcare Corporation Deerfield, IL 60015 USA) or Surgiflo® (Ethicon Incorporated, Somerville, NJ) [14].

Randomization occurred, intraoperatively (once it was clear the patient had met the intra-operative inclusion/exclusion criteria) on a 1:1 basis and was assigned by an electronic randomization system.

### Follow-up visits

Patients were approached for consent and were screened up to 2 months prior to enrollment and reviewed pre-operatively, intra-operatively, 24 hours post operatively, prior to discharge (if the subject was discharged within 24 hours of surgery, then only discharge data was required) and at 30 days after their procedure (between day 21 and day 35).

Visits conducted outside this window were considered non-compliant and a protocol deviation.

### Outcomes

The primary effectiveness endpoint of the study was the achievement of hemostasis (yes/no) within 10 minutes. Time to hemostasis was also assessed at 1, 2, 3, 5, 7, and 10 minutes. The primary safety endpoint of the study was the absence of proven infection and bleeding related adverse events (AEs).

The secondary effectiveness endpoint was the absence of reintervention for post- operative bleeding. Other secondary outcomes evaluated in this study included duration of the surgical procedure; incidence of postoperative hematoma; duration of drainage, drainage volume; postoperative pain measured by a visual analogue scale (VAS); and incidence of AEs.

### Definitions

Hemostasis was defined as the absence of any blood egressing from the applied hemostatic agent, or from the diathermy site if no topical agent was used, within 10 minutes.

Bleeding was classified as Oozing (The rate of bleeding was slow. This was what one may expect from capillary, venular, or arteriolar bleeding); Mild (The rate of bleeding was slightly faster than oozing. There was no pulsatile flow present. This was what one may expect from capillary, venular, or arteriolar bleeding); Moderate (There might be a weak pulsatile flow present. If there was a pulsatile flow, the rate of blood flow was like the rate of flow for mild bleeding. If there was not pulsatile flow, the rate of blood flow was faster than mild bleeding); and Severe (for the most part, pulsatile flow was present. If there was no pulsatile flow, then the rate of blood flow was extremely rapid).

Pain scores were measured post-operatively using the visual analogue scale (VAS) (100 mm scale). Subjects were asked to rate their pain on the scale from no pain to worst possible pain. If the pain score was taken via telephone, subjects were asked to rate their pain between 0 and 10 with no pain being 0 and the worst pain possible being 10.

### Statistical analysis

The protocol estimated that approximately 90 patients should be enrolled.

The Intent-to-Treat (ITT) population included all randomized subjects, regardless of treatment received. The Per Protocol (PP) population included all subjects, who were randomized and treated with AMP-SPH, and had no major protocol deviations. Additionally, "As Treated" (AT) population (subjects assessed according to the treatment received rather than treatment assigned) was evaluated.

Mean and standard deviation (SD); mean and 95% confidence interval (95% CI); median and interquartile range (IqR), and number (percentage) were used as appropriate.

### Results

#### Preoperative demographic and clinical characteristics

A total of 90 patients were enrolled across two study sites between November 2015 and July 2017. Screening concluded in July 2017 upon reaching the target enrollment. The study duration was 21 months, from the first patient's initial visit to the last patient's final visit. Of the 90 enrolled participants, 44 (48.9%) were assigned to the AMP group and 46 (51.1%) to the SC group (Fig 1).

In the overall study sample, mean age was 57.2 ± 13.5 years, with no significant differences between AMP (55.7 ± 14.7 years) and SC (58.6 ± 12.2 years) groups (mean difference: −2.9 ± 13.5 years, 95%CI: −8.6 to 2.8; p.310).

Table 1 summarizes the main demographic characteristics of the study sample.

There was no statistical difference between the height, weight, or body mass index (BMI) between groups.

The proportion of patients with a history of menorrhagia (32.7% versus 28.3%, p=.373) or incidence of dysmenorrhea (18.2% versus 11.1%, p=.343) was greater in the AMP group than in the SC group, although such differences were not statistically significant. Women treated for malign pathology were 61.4% (27/44) and 78.3% (36/46) in the AMP and SC groups, respectively. There were no significant differences between groups in any of the preoperative demographic and clinical characteristics (Table 1).

#### Surgical procedures and type of surgery

There were no significant differences between study groups in either indication of surgical procedure or type of surgical procedures (Table 2).

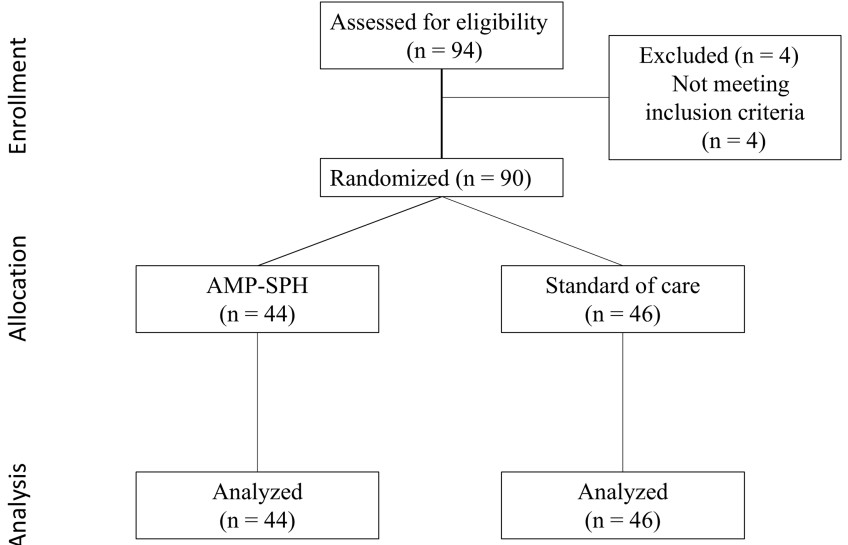

**Fig 1. CONSORT diagram showing the flow of participants through each stage of this randomized trial.** AMP-SPH: Absorbable modified polymer starch powder hemostat.

**Table 1. Overview of the main preoperative demographic and clinical characteristics of the study population.**

|  | Overall (n = 90) | AMP (n = 44) | SC (n = 46) | Mean difference (95%CI) | P value |
|---|---|---|---|---|---|
| Age, years Mean±SD | 57.2 ± 13.5 | 55.7 ± 14.7 | 58.6 ± 12.2 | −2.9 (−8.6 to 2.8) | 0.2784[a] |
| Race, % (n/N) Caucasian | 96.7% (87/90) | 97.7% (43/44) | 95.7% (44/46) | 2.1% (−5.3% to 9.4%) | 0.6821[b] |
| BMI, Kg/m² Mean±SD | 26.8 ± 5.7 | 27.7 ± 5.7 | 26.1 ± 5.7 | 1.6 (−0.8 to 4.0) | 0.1792[a] |
| Temperature, ºC Mean±SD | 36.5 ± 0.4 (n = 74) | 36.5 ± 0.4 (n = 37) | 36.4 ± 0.4 (n = 37) | 0.1 (−0.1 to 0.2) | 0.2858[a] |
| SBP, mmHg Mean±SD | 123.8 ± 18.2 (n = 74) | 125.1 ± 18.5 (n = 37) | 122.6 ± 18.2 (n = 37) | 2.5 (−6.0 to 11.0) | 0.5597[a] |
| DBP, mmHg Mean±SD | 74.0 ± 11.3 (n = 74) | 75.1 ± 9.9 (n = 37) | 73.0 ± 12.6 (n = 37) | 2.1 (−3.2 to 7.4) | 0.4280[a] |
| HR, bpm Mean±SD | 77.8 ± 10.0 (n = 72) | 75.8 ± 9.6 (n = 37) | 80.0 ± 10.0 (n = 35) | −4.2 (−8.8 to 0.4) | 0.0694[a] |
| Preop INR Mean±SD | 0.98 ± 0.06 (n = 51) | 0.98 ± 0.06 (n = 22) | 0.97 ± 0.06 (n = 29) | 0.01 (−0.03 to 0.04) | 0.5582[a] |
| Use of Blood Modifiers, % (n/N) | 7.8% (7/90) | 4.6% (2/44) | 10.9% (5/46) | −6.3% (−17.2% to 4.6%) | 0.2685[b] |
| Hypertension, % (n/N) | 30.0% (27/90) | 29.6% (13/44) | 30.4% (14/46) | −0.9% (−19.8% to 18.0%) | 0.9344[b] |
| Tachycardia, % (n/N) | 2.2% (2/90) | 0.0% (0/44) | 4.4% (2/46) | −4.3% (−10.2% to 1.5%) | 0.1617[b] |
| Menorrhagia, % (n/N) | 32.6% (29/89) | 37.2% (16/43) | 28.3% (13/46) | 8.9% (−10.5% to 28.4%) | 0.3734[b] |
| Dysmenorrhea, % (n/N) | 14.6% (13/89) | 18.2% (8/44) | 11.1% (5/45) | 7.1% (−7.6% to 21.7%) | 0.3426[b] |
| Malignancy, % (n/N) | 70.0% (63/90) | 61.4% (27/44) | 78.3% (36/46) | −16.9% (−35.6% to 1.8%) | 0.0819[b] |
| History of Endometrial Disease, % (n/N) | 5.6% (5/90) | 4.6% (2/44) | 6.5% (3/46) | −2.0% (−11.4% to 7.4%) | 0.6961[b] |
| Uterine Fibroids, % (n/N) | 39.3% (35/89) | 44.2% (19/43) | 34.8% (16/46) | 9.4% (−10.8% to 29.6%) | 0.3670[b] |

a Independent samples t Student test.

b Chi-squared test.

AMP: Absorbable modified polymer starch powder hemostat; SC: Standard care; SD: Standard deviation; 95%CI: 95% Confidence interval; BMI: Body mass index; SBP: Systolic blood pressure; DBP: Diastolic blood pressure; HR: heart rate; bpm: Beats per minute; INR: International normalized ratio; NA: Not applicable.

**Table 2. Overview of the surgical procedure and type of surgery among the study population.**

| Indication*, % (n/N) | Overall (N = 90) | AMP Group (N = 44) | SC Group (N = 46) | Mean difference (95%CI) | P value[a] |
|---|---|---|---|---|---|
| Hysterectomy | 81.1% (73/90) | 79.6% (35/44) | 82.6% (38/46) | −3.1% (−19.3% to 13.1%) | 0.7176 |
| Endometrial Resection | 0.0% (0/90) | 0.0% (0/44) | 0.0% (0/46) | 0.0% (N/A) | N.A. |
| Cystectomy | 4.4% (4/90) | 6.8% (3/44) | 2.2% (1/46) | 4.6% (−3.9% to 13.2%) | 0.2927 |
| Myomectomy | 4.4% (4/90) | 6.8% (3/44) | 2.2% (1/46) | 4.6% (−3.9% to 13.2%) | 0.2927 |
| Other | 77.8% (70/90) | 72.7% (32/44) | 82.6% (38/46) | −9.9% (−27.0% to 7.2%) | 0.2616 |
| Lymphadenectomy | 36.7% (33/90) | 36.4% (16/44) | 37.0% (17/46) | −0.6% (−20.5% to 19.3%) | 0.9532 |
| **Type of Surgery, (n/N)** | **Overall (N = 90)** | **AMP Group (N = 44)** | **SC Group (N = 46)** | **Mean difference (95%CI)** | |
| Open | 46.7% (42/90) | 47.7% (21/44) | 45.7% (21/46) | 2.1% (−18.5% to 22.7%) | 0.8501 |
| Laparoscopic | 46.7% (42/90) | 45.5% (20/44) | 47.8% (22/46) | −2.4% (−23.0% to 18.2%) | 0.8279 |
| Other# | 6.7% (6/90) | 6.8% (3/44) | 6.5% (3/46) | 0.3% (−10.0% to 10.6%) | 0.9547 |

*Not mutually exclusive. Most patients had concomitant procedures along with, e.g., hysterectomy. #E.g.: robotic, vaginal, etc.

a Chi-squared test.

AMP: Absorbable modified polymer starch powder hemostat; SC: Standard care: SD: Standard deviation; 95%CI: 95% Confidence interval; NA: Not applicable.

Most patients enrolled (81.1%) required a hysterectomy with or without a concomitant procedure. In the overall study population, open surgery was performed in 42 (46.7%) subjects, a laparoscopic approach was performed in 42 (46.7%) patients, and 6 (6.7%) patients underwent robotic procedures.

## Primary and secondary effectiveness outcomes

Most of the subjects experienced oozing (37.8%; 34/90) or mild (47.8%; 43/90) bleeding, with no significant differences between AMP (36.4% and 52.3%, respectively) and SC (39.1% and 43.5%, respectively) groups, p = .649. Twelve (13.3%) patients had moderate bleeding, 4 (9.1%) in the AMP group and 8 (17.4%) in the SC group (p = .250). Only 1 (1.1%) patient, who was assigned to the AMP group, had severe bleeding.

The primary effectiveness endpoint was the achievement of hemostasis, as defined as the absence of any blood egressing from the applied hemostatic agent or from the diathermy site (if no topical agent was used) within 10 minutes of application.

There was no statistically significant difference between the two groups with a slightly higher rate of achieved hemostasis in the AMP group compared to SC group (97.7% (43/44) VS 93.5% (43/46); Mean difference: 4.2%; 95%CI: −4.1% to 12.6%; p = .337.

An overview of the proportion of patients who achieved hemostasis in the ITT, PP, and AT populations is shown in Table 3.

**Table 3. Proportion of patients who achieved hemostasis.**

| | AMP (n = 44) | SC (n = 46) | Mean difference (95%CI) | P value[a] |
|---|---|---|---|---|
| ITT, % (n/N) | 97.7% (43/44) | 93.5% (43/46) | 4.2% (−4.1% to 12.6%) | 0.3366 |
| AT, % (n/N) | 97.7% (43/44) | 93.5% (43/46) | 4.2% (−4.1% to 12.6%) | 0.3366 |
| PP, % (n/N) | 95.5% (21/22) | 96.2% (25/26) | −0.7% (−12.1% to 10.7%) | 0.8684 |

[a]Chi-squared test.

AMP: Absorbable modified polymer starch powder hemostat; SC: Standard care; 95%CI: 95% Confidence interval; ITT: Intent-to-treat; AT: As Treated; PP: Per protocol.

The time required to achieve hemostasis was lower in the AMP (1.91±1.15 minutes) versus the SC (2.28±2.09 minutes) group, although this difference was not statistically significant (mean difference: −0.37±1.68 minutes; 95%CI: −1.09 to 0.35; p=.309).

One patient was randomized to usual care and did not achieve hemostasis within 10 minutes, but subsequently had AMP-SPH applied as a secondary measure to cease bleeding. Other hemostats were only used in 20.4% of SC group patients. FloSeal® was used in 13.6% of procedures and Surgiflo® in 6.8% of procedures.

The total time of surgery was 204.8±109.9 and 216.7±114.4 minutes in AMP and SC groups, respectively, without a significant difference between groups (mean difference: −11.9±112.2 minutes; 95%CI: −58.9 to 35.1 minutes; p=.616). During the procedure, red blood cell units were given to a total of 9 (10.2%) patients. More patients in the standard care arm required blood products (17.9%) compared to those in the AMP group (4.8%), although this difference was not statistically significant (p=.055).

During this study, no patients required reintervention for post-operative bleeding.

## Safety outcomes

The AMP-SPH was applied to the pelvis or pelvic floor in 30 patients (69.7%). A full list of application sites can be found in S1 Table.

The most commonly reported AE in the AMP group was hematoma, which occurred in 5 cases: 2 cases of pelvic hematoma (one with vaginal spotting); 1 Douglas hematoma; 1 vaginal cuff hematoma; and 1 cupula hematoma. There was one occurrence of hematoma in the SC group.

A total of 29 AEs (24 non-serious and 5 serious AEs) were reported, 12 AEs in the AMP group and 17 in the SC group (Table 4).

Detailed information of the different AEs, both non-serious and serious, are shown in S2, S3, and S4 Tables, respectively.

At 24 hours postoperatively, pain was 4.04±2.19 and 4.64±2.21 in the AMP and SC groups, respectively (mean difference: −0.60±2.22; 95%CI: −1.61 to 0.44; p=.254). Pain at discharge was 2.36±1.13 and 2.53±1.6 in the AMP and SC groups, respectively (−0.17±1.40; 95%CI: −0.84 to 0.50; p=.614). At Day 30 postoperatively, pain was 1.04±0.96 and 0.98±0.96 in the AMP and SC groups, respectively (mean difference: 0.06±0.97, 95%CI: −0.42 to 0.54; p=.804).

At 24 hours postoperatively, a greater proportion of subjects in the SC group had a drain in situ (incidence rate difference: 11.2%; 95%CI: −17.7% to 40.1%; p=.455), although such a difference was not statistically significant. Nine (19.6%) and 3 (6.8%) patients in the SC and AMP groups, respectively, had a drain at discharge (incidence rate difference: 12.8%; 95%CI: −2.3% to 27.8%; p=.108).

Although drainage volume was lower in the AMP group at both 24 hours postoperatively (mean difference: −74.4±285.9 mL; 95%CI: −250.0 to 01.2 mL; p=.397) and discharge (mean difference: −544.0±753.1; 95%CI: −1662.7 to 574.7; p=.304), such differences were not statistically significant.

**Table 4. Overview of the adverse events (AEs) classification.**

| | Events (Subjects, %) | | | | |
| --- | --- | --- | --- | --- | --- |
| | Overall n=90) | AMP (n=44) | SC (n=46) | Mean difference (95% CI) | P value[a] |
| Non-Serious AE | 19 (21.1%) | 8 (18.2%) | 11 (23.9%) | −5.7% (−22.5%, 11.1%) | 0.5101 |
| Non-Serious Device Related AE | 3 (3.3%) | 3 (6.8%) | 0 (0, 0.0%) | 6.8% (−0.6%, 14.3%) | 0.0737 |
| Serious AE | 5 (5.6%) | 2 (4.5%) | 3 (6.5%) | −2.0% (−11.4%, 7.4%) | 0.6797 |
| Serious Device Related AE | 1 (1.1%) | 1 (2.3%) | 0 (0, 0.0%) | 2.3% (−2.1%, 6.7%) | 0.3036 |

[a]Chi-squared test.

AMP: Absorbable modified polymer starch powder hemostat; SC: Standard care; AE: Adverse event.

In the AMP-SPH group, hemostasis within 10 minutes was achieved in all but one patient; bleeding in this case was subsequently controlled by unspecified methods. In the SC group, three patients failed to achieve hemostasis within 10 minutes. Bleeding was controlled in these cases using AMP-SPH (n = 1), sutures (n = 1), and unspecified methods (n = 1). No intraoperative rebleeding events or postoperative reinterventions for bleeding were reported during the study.

## Discussion

Hemorrhage is a rare but serious potential complication of gynecological surgery, and postoperative bleeding may occur despite meticulous surgical technique [15,16]. Indeed, acute postoperative hemorrhage is the most frequent cause of return to the operating theatre [16].

The capacity to improve hemostasis in surgical procedures is a key factor for preventing blood loss, reducing perioperative morbidity and surgery times, and improving surgical outcomes [17].

While hemostasis is typically achieved through suturing, electrocautery, or surgical clips, several adjunctive hemostatic agents have been developed for use over the last two decades [10–12]. For instance, the use of local hemostatic agents can be considered an addition to traditional surgical coagulation and ligation in gynecologic oncology to obtain adequate hemostasis [18].

In general terms, hemostatic agents can be divided into active and passive agents, with both groups having utility in different procedures as adjunctive therapies for controlling surgical bleeding [19].

Active hemostatic agents work at the end or independent of the coagulation cascade and thus also work in patients with compromised coagulation [20–22]. These agents can be used effectively in patients with spontaneous or drug-induced coagulation disorders and are effective for a wide range of bleeding grades (including pulsatile arterial bleeding) [20–24].

Passive hemostatic agents generally provide a structure for platelets to aggregate and activate, while forming a matrix at the site of bleeding, which then allows the coagulation cascade to be activated through contact activation, finally leading to the formation of a fibrin clot [20–22]. The efficacy of passive agents depends on the patient's own endogenous presence of clotting factors that result in fibrin production to achieve hemostasis, so they are only effective in patients who have an intact coagulation system [20–22,25–28]. These hemostatic agents are most effective when used in areas where the bleeding is of a lower grade, or to treat large oozing surfaces [9–12,29].

The AMP-SPH has been developed for use in surgical procedures or injuries as an adjunct hemostat when control of bleeding from capillary, venous or arteriolar vessels by pressure, ligature or other conventional means is either ineffective or impractical.

The current study compared the effectiveness and safety of an AMP-SPH to usual care when used in adult subjects during open or laparoscopic gynecological procedures.

The AMP-SPH alone was not inferior to standard care, both in the proportion of subjects who achieved hemostasis within 10 minutes (primary endpoint) and in every one of the secondary effectiveness endpoints. In addition, it is important to consider that, within the standard care group, the use of other hemostatic agents was allowed, and active hemostatic products were used in 20.4% of cases.

Regarding safety, the incidence of AEs, both non-serious and serious, was similar in both groups.

The most commonly reported AE in the AMP group was hematoma, which occurred in 5 cases, while there was one case of hematoma in the SC group.

The incidence of hematoma formation following pelvic gynecological surgery is highly variable, ranging from 25% to 98% [30]. Established risk factors include the type of surgery, patient comorbidities, surgical technique, oncologic resections, difficult anatomical exposure, prior surgeries, recurrent tumors, and previous radiation therapy [31]. Although these factors may have influenced the observed, non-significant differences in hematoma rates between the study groups, no differences in the distribution of these risk factors were identified.

 

Infection was reported in 4 cases, two patients in each group. Although a higher proportion of patients in the SC group (17.9%) was observed to require blood products compared to the AMP group (4.8%), this difference did not reach statistical significance. Even though cost analysis was beyond the scope of the current study, the finding may suggest a potential reduction in procedural costs.

There are some limitations that need to be taken into consideration when interpreting the results of this study, the most important of which was that standard care strategy was at the investigator's discretion. In fact, in addition to the usual hemostatic techniques, the SC group could receive two commonly used gelatin-thrombin-based flowable active hemostatic agents (FloSeal® and Surgiflo®).

This fact could have skewed the results in favor of the SC group.

Another important limitation of this study is the lack of a statistically valid sample size justification. While a prospective sample size calculation would have been the ideal approach, a post-hoc analysis was performed to estimate the number of patients required to detect significant differences based on the observed results. With an alpha risk of 0.05 in a two-sided test and sample sizes of 44 patients in the AMP group and 46 in the SC group, the statistical power to detect the observed differences as significant was only 16%. Furthermore, to achieve a power of 80% with an alpha of 0.05, a total of 353 patients per group would have been required for the observed differences in the proportion of patients achieving hemostasis within 10 minutes to reach statistical significance.

Additionally, the absence of a multivariate analysis to assess potential covariate associations with hemostasis, which could have offered a more detailed statistical evaluation, may be considered a limitation of the study.

Finally, the current study did not evaluate the cost-effectiveness of both hemostatic strategies, which could have provided additional context to the findings.

## Conclusions

The AMP-SPH product was safe and effective and not inferior to standard care in the control of lower grade bleeding for patients undergoing gynecology procedures and the cessation of bleeding tended to be faster with the use of AMP-SPH than with standard methods.

Based on these findings, future studies are warranted to analyze the cost-effectiveness of the use of hemostatic agents in gynecological surgery, in addition to head-to-head comparisons between different hemostatic agents.

## Clinical trial registration

NCT02835391. Registered July 31, 2015.

## Supporting information

**S1 Table. Sites of application of the absorbable modified polymer (AMP) starch powder hemostat.**
(DOCX)

**S2 Table. Overview of the different non-serious adverse events reported in the study.**
(DOCX)

**S3 Table. Overview of the Serious adverse events documented over the study follow-up.**
(DOCX)

**S4 Table. Time of onset and evolution of the different serious adverse events reported during the study.**
(DOCX)

## Acknowledgments

The writing up and formatting of the manuscript by Ciencia y Deporte S.L. is greatly acknowledged. Alberto Izarra and Rafella De Santis, both Baxter Medical Affairs, supported in data analysis and coordinating the author collaboration on this manuscript.

## Author contributions

**Conceptualization:** Amparo Garcia-Tejedor, Jordi Ponce, Serena Cappuccio, Giovanni Scambia.

**Data curation:** Marc Barahona , Serena Cappuccio, Barbara Costantini, Camilla Fedele, Lola Marti-Cardona.

**Formal analysis:** Amparo Garcia-Tejedor, Jordi Ponce, Marc Barahona , Barbara Costantini.

**Funding acquisition:** Amparo Garcia-Tejedor.

**Investigation:** Amparo Garcia-Tejedor, Jordi Ponce, Marc Barahona , Serena Cappuccio, Barbara Costantini, Camilla Fedele, Lola Marti-Cardona, Giovanni Scambia.

**Methodology:** Amparo Garcia-Tejedor, Jordi Ponce, Giovanni Scambia.

**Project administration:** Marc Barahona , Barbara Costantini, Lola Marti-Cardona.

**Resources:** Serena Cappuccio.

**Software:** Lola Marti-Cardona.

**Supervision:** Giovanni Scambia.

**Validation:** Serena Cappuccio, Barbara Costantini, Camilla Fedele.

**Visualization:** Serena Cappuccio.

**Writing – original draft:** Marc Barahona , Camilla Fedele.

**Writing – review & editing:** Amparo Garcia-Tejedor, Jordi Ponce, Giovanni Scambia.

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
