## [Decision Letter · Decision Letter 0]

21 Mar 2025

PONE-D-25-02293Effectiveness and safety of an absorbable modified polymer starch powder hemostat versus usual care in gynecology procedures: A prospective, multi-center, and randomized study.PLOS ONE

Dear Dr. Garcia-Tejedor,

Thank you for submitting your manuscript to PLOS ONE. After careful consideration, we feel that it has merit but does not fully meet PLOS ONE’s publication criteria as it currently stands. Therefore, we invite you to submit a revised version of the manuscript that addresses the points raised during the review process.

We look forward to receiving your revised manuscript.

Kind regards,

Hideto Sano

Academic Editor

PLOS ONE

 [Cryolife, Europe

Baxter (Founded Medical writing)]. 

[Barbara Costantini, Jordi Ponce and Giovanni Scambia served as speakers for Baxter and have received speaker honoraria according to fair market value.].

5. We note that you have indicated that there are restrictions to data sharing for this study. For studies involving human research participant data or other sensitive data, we encourage authors to share de-identified or anonymized data. However, when data cannot be publicly shared for ethical reasons, we allow authors to make their data sets available upon request. For information on unacceptable data access restrictions, please see http://journals.plos.org/plosone/s/data-availability#loc-unacceptable-data-access-restrictions.

Reviewers' comments:

Reviewer's Responses to Questions

**Comments to the Author**

1. Is the manuscript technically sound, and do the data support the conclusions?

Reviewer #1: No

Reviewer #2: Yes

2. Has the statistical analysis been performed appropriately and rigorously? 

Reviewer #1: No

Reviewer #2: Yes

3. Have the authors made all data underlying the findings in their manuscript fully available?

Reviewer #1: Yes

Reviewer #2: Yes

4. Is the manuscript presented in an intelligible fashion and written in standard English?

Reviewer #1: Yes

Reviewer #2: Yes

5. Review Comments to the Author

Reviewer #1: The study lacks a statistically valid sample size justification. That could not be found in the protocol, either. The paper mentions that 90 subjects are needed as per the protocol. How was that calculated? This reviewer found no valid statistical design section in the protocol unless it is under another section. One should not have to search for this information. Section 9.0 (Statistical Considerations) of the protocol is totally inadequate. All it mentions is an ITT and per protocol strategy. Where or what was the intended non inferiority margin? Since non inferiority was mentioned in the manuscript, it should have been incorporated into the design of the study.

Apparently the ITT and per protocol results compare as they are the same as noted in Table 3. However, even though the groups look fairly balanced was there any attempt to look at covariate association for achieving hemostasis along with treatment in a multivariate setting such as the logistic regression? The investigation does seem incomplete, unless other possible associative factors with hemostasis are examined in a more rigorous statistical setting. The statistical handling of this data (design and analysis) is obviously incomplete.

Reviewer #2: I found the manuscript, to be of high quality. The methodology is sound, the data are presented clearly, and the discussion is insightful. The authors have effectively addressed the research question, and the article is well-designed for publication

6. PLOS authors have the option to publish the peer review history of their article (what does this mean? ). If published, this will include your full peer review and any attached files.

**Do you want your identity to be public for this peer review?** For information about this choice, including consent withdrawal, please see our Privacy Policy .

Reviewer #1: No

Reviewer #2: **Yes: ** Selim Misirlioglu

---

## [Author Response · Author response to Decision Letter 1]

25 Apr 2025

Response to the reviewers, manuscript PONE-D-25-02293, entitled

“Effectiveness and safety of an absorbable modified polymer starch powder hemostat versus usual care in gynecology procedures: A prospective, multi-center, and randomized study”.

Dear Editor-in-Chief, I have considered the comments made by the reviewers and I hope that the paper improves greatly thanks to these comments.

I am sending you the revised manuscript and the rebuttal letter providing a point-by-point response to each of the numbered reviewer comments.

#Journal requirements:

It was corrected.

No specific numbers have been assigned to the Grants received.

[Cryolife, Europe

Baxter (Founded Medical writing)].

Thank you very much indeed for the comment, we highly appreciate it.

The following sentence was added in the Funding section and in the cover letter: “Cryolife was involved in study funding, study design, provision of study materials, data collection and analysis”.

“Baxter was involved in medical writing funding, preparation of the manuscript, decision to publish”.

[Barbara Costantini, Jordi Ponce and Giovanni Scambia served as speakers for Baxter and have received speaker honoraria according to fair market value.].

The following sentence was added in the “Conflict of Interests” section and in the cover letter: “This does not alter our adherence to PLOS ONE policies on sharing data and materials.”

5. We note that you have indicated that there are restrictions to data sharing for this study. For studies involving human research participant data or other sensitive data, we encourage authors to share de-identified or anonymized data. However, when data cannot be publicly shared for ethical reasons, we allow authors to make their data sets available upon request. For information on unacceptable data access restrictions, please see http://journals.plos.org/plosone/s/data-availability#loc-unacceptable-data-access-restrictions.

The following sentence was added in the Data Availability Statement: “Our study utilizes third-party data that are the property of Artivion, formerly known as CryoLife Europa. As these data are proprietary, they cannot be publicly shared. However, interested researchers may contact Artivion directly to inquire about potential access to the data (https://artivion.com/; 1655 Roberts Blvd., NW; Kennesaw, GA 30144 USA.”.

It was corrected.

Reviewers' comments:

Reviewer's Responses to Questions

Comments to the Author

1. Is the manuscript technically sound, and do the data support the conclusions?

Reviewer #1: No

Reviewer #2: Yes

2. Has the statistical analysis been performed appropriately and rigorously?

Reviewer #1: No

Reviewer #2: Yes

3. Have the authors made all data underlying the findings in their manuscript fully available?

Reviewer #1: Yes

Reviewer #2: Yes

4. Is the manuscript presented in an intelligible fashion and written in standard English?

Reviewer #1: Yes

Reviewer #2: Yes

5. Review Comments to the Author

Please use the space provided to explain your answers to the questions above. You may also include additional comments for the author, including concerns about dual publication, research ethics, or publication ethics. (Please upload your review as an attachment if it exceeds 20,000 characters).

Reviewer #1:

1. The study lacks a statistically valid sample size justification. That could not be found in the protocol, either. The paper mentions that 90 subjects are needed as per the protocol. How was that calculated? This reviewer found no valid statistical design section in the protocol unless it is under another section. One should not have to search for this information. Section 9.0 (Statistical Considerations) of the protocol is totally inadequate. All it mentions is an ITT and per protocol strategy. Where or what was the intended non inferiority margin? Since non inferiority was mentioned in the manuscript, it should have been incorporated into the design of the study.

Thank you very much indeed for the comment, we highly appreciate it.

We fully agree with the reviewer about the limitation of the lack of statistically valid sample size justification. This was added as a study limitation.

The current study was a Phase IV, observational (post-market) study designed to provide descriptive and exploratory data rather than confirmatory evidence. As stated in the study protocol, the sample size was not determined to achieve a specific statistical power to detect differences between the PerClot and Standard of Care (SoC) patient populations. Instead, this study followed an observational, case-controlled design, in which patients were grouped based on treatment exposure rather than through true randomization, which is characteristic of experimental studies such as randomized controlled trials (RCTs). The PerClot group represented the case study cohort, while the SoC group served as the control, allowing for comparative assessment of specific clinical characteristics between the two cohorts.

Given the exploratory nature of this study, our primary objective was to gather preliminary evidence rather than to definitively establish non-inferiority. Although prospective sample size calculation is the preferred methodological approach, as the reviewer correctly noted, a post-hoc analysis was conducted to estimate the number of patients required to detect significant differences based on the observed results. This analysis provided valuable insights into the potential adequacy of the study’s sample size, reinforcing the descriptive and hypothesis-generating role of this investigation.

The following paragraph was added as a study limitation: “Another important limitation of this study is the lack of a statistically valid sample size justification. While a prospective sample size calculation would have been the ideal approach, a post-hoc analysis was performed to estimate the number of patients required to detect significant differences based on the observed results. With an alpha risk of 0.05 in a two-sided test and sample sizes of 44 patients in the AMP group and 46 in the SC group, the statistical power to detect the observed differences as significant was only 16%. Furthermore, to achieve a power of 80% with an alpha of 0.05, a total of 353 patients per group would have been required for the observed differences in the proportion of patients achieving hemostasis within 10 minutes to reach statistical significance”.

2. Apparently the ITT and per protocol results compare as they are the same as noted in Table 3. However, even though the groups look fairly balanced was there any attempt to look at covariate association for achieving hemostasis along with treatment in a multivariate setting such as the logistic regression? The investigation does seem incomplete, unless other possible associative factors with hemostasis are examined in a more rigorous statistical setting. The statistical handling of this data (design and analysis) is obviously incomplete.

Thank you for your insightful comment. We fully acknowledge the importance of examining potential covariate associations with hemostasis in a more rigorous statistical framework, such as logistic regression, to strengthen the analysis. Unfortunately, we do not have access to the original dataset but only to the statistical report provided by the study promoter, which limits our ability to perform additional multivariate analyses. Consequently, we are unable to explore potential confounding factors that may influence hemostasis outcomes. While our study primarily focused on comparing ITT and per-protocol results, we recognize that a more comprehensive statistical approach could provide further insights. We appreciate your valuable suggestion and acknowledge this as a limitation of our study.

The following sentence was added as a limitation of the study: “Additionally, the absence of a multivariate analysis to assess potential covariate associations with hemostasis, which could have offered a more detailed statistical evaluation, may be considered a limitation of the study”.

Reviewer #2:

I found the manuscript, to be of high quality. The methodology is sound, the data are presented clearly, and the discussion is insightful. The authors have effectively addressed the research question, and the article is well-designed for publication.

Thank you very much indeed for this insightful comment; we truly appreciate it.

---

## [Decision Letter · Decision Letter 1]

19 Jun 2025

PONE-D-25-02293R1Effectiveness and safety of an absorbable modified polymer starch powder hemostat versus usual care in gynecology procedures: A prospective, multi-center, and randomized study.PLOS ONE

Dear Dr. Garcia-Tejedor,

Thank you for submitting your manuscript to PLOS ONE. After careful consideration, we feel that it has merit but does not fully meet PLOS ONE’s publication criteria as it currently stands. Therefore, we invite you to submit a revised version of the manuscript that addresses the points raised during the review process.

We look forward to receiving your revised manuscript.

Kind regards,

Hideto Sano

Academic Editor

PLOS ONE

Journal Requirements:

Reviewers' comments:

Reviewer's Responses to Questions

**Comments to the Author**

1. If the authors have adequately addressed your comments raised in a previous round of review and you feel that this manuscript is now acceptable for publication, you may indicate that here to bypass the “Comments to the Author” section, enter your conflict of interest statement in the “Confidential to Editor” section, and submit your "Accept" recommendation.

Reviewer #1: All comments have been addressed

Reviewer #3: (No Response)

Reviewer #4: All comments have been addressed

2. Is the manuscript technically sound, and do the data support the conclusions?

Reviewer #1: (No Response)

Reviewer #3: No

Reviewer #4: Yes

3. Has the statistical analysis been performed appropriately and rigorously? 

Reviewer #1: (No Response)

Reviewer #3: No

Reviewer #4: N/A

4. Have the authors made all data underlying the findings in their manuscript fully available?

Reviewer #1: (No Response)

Reviewer #3: No

Reviewer #4: Yes

5. Is the manuscript presented in an intelligible fashion and written in standard English?

Reviewer #1: (No Response)

Reviewer #3: Yes

Reviewer #4: Yes

6. Review Comments to the Author

Reviewer #1: (No Response)

Reviewer #3: In this revised submission, Dr. Garcia-Tejedor and colleagues compare the use of absorbable modified polymer starch powder hemostat (AMP-SPH) compared to standard of care (SC). The submission could be improved with respect to technique, data, and statistics by the following:

1. The bleeding severity descriptions provided suggest that AMP-SPH may have been used outside of its approved labeling. Providing an explanation of why this was permitted would be useful.

2. Achieving hemostasis at 10 minutes was the primary endpoint, but 10 minutes is considered long for the primary endpoint in time to hemostasis studies in the current environment. Supplying a table with the results for the degree of hemostasis at all observed time points would be helpful.

3. The bleeding severity score use was only semi-quantitative. Providing a discussion of why a quantitative bleeding severity score was not employed, what such scores exist, and how not using one may effect results would be appropriate.

4. Addition to the discussion of how electrocautery was the SC when AMP-SPH is approved for use when conventional treatments are ineffective or impractical. Explaining why it would not have been more appropriate to use another established hemostat (for example a flowable) as the SC in all SC subjects should be discussed.

5. Reporting of the incidence of rebleeding after 10 minutes intraoperatively.

6. Explanation of what were the pre-analysis criteria for statistical non-inferiority.

7. Clarification as to why all subjects were not able to be included in the final analysis.

8. Discussion of why there was a trend toward more hematomas in the AMP-SPH group.

9. In the abstract as well as on page 2 line 44-45 and page 16, line 357-9, the word “observed” could be added for the comparison on amount of blood transfusions being less in the AMP-SPH group as this difference was not statistically significant.

Reviewer #4: The authors of the manuscript have properly addresed all the Reviewer´s comments and concerns. Congratulations.

7. PLOS authors have the option to publish the peer review history of their article (what does this mean? ). If published, this will include your full peer review and any attached files.

**Do you want your identity to be public for this peer review?** For information about this choice, including consent withdrawal, please see our Privacy Policy .

Reviewer #1: No

Reviewer #3: No

Reviewer #4: No

---

## [Author Response · Author response to Decision Letter 2]

8 Jul 2025

Response to the reviewers, manuscript PONE-D-25-02293R1, entitled

“Effectiveness and safety of an absorbable modified polymer starch powder hemostat versus usual care in gynecology procedures: A prospective, multi-center, and randomized study”.

Dear Editor-in-Chief, I have considered the comments made by the reviewers and I hope that the paper improves greatly thanks to these comments.

I am sending you the revised manuscript and the rebuttal letter providing a point-by-point response to each of the numbered reviewer comments.

# Journal Requirements:

Reference list has been reviewed.

# Review Comments to the Author

Reviewer #1: (No Response)

Reviewer #3: In this revised submission, Dr. Garcia-Tejedor and colleagues compare the use of absorbable modified polymer starch powder hemostat (AMP-SPH) compared to standard of care (SC). The submission could be improved with respect to technique, data, and statistics by the following:

1. The bleeding severity descriptions provided suggest that AMP-SPH may have been used outside of its approved labeling. Providing an explanation of why this was permitted would be useful.

According to the study protocol, and the intraoperative inclusion criteria: Subject experiences generalized oozing or mild to moderate bleeding which is consistent with the requirement for a hemostatic agent. Bleeding was defined and documented in the CRF as:

• Oozing - The rate of bleeding is slow. This is what one may expect from capillary, venular, or arteriolar bleeding.

• Mild bleeding - The rate of bleeding is slightly faster than oozing. There is no pulsatile flow present. This is what one may expect from capillary, venular, or arteriolar bleeding.

• Moderate bleeding. There may be a weak pulsatile flow present. If there is a pulsatile flow, the rate of blood flow is similar to the rate of flow for mild bleeding. If there is no pulsatile flow, the rate of blood flow is faster than mild bleeding.

• Severe bleeding. For the most part, pulsatile flow is present. If there is no pulsatile flow, then the rate of blood flow is extremely rapid.

If we were to apply the Vibe Scale (Lewis KM, Li Q, Jones DS, Corrales JD, Du H, Spiess PE, Lo Menzo E, DeAnda A Jr. Development and validation of an intraoperative bleeding severity scale for use in clinical studies of hemostatic agents. Surgery. 2017 Mar;161(3):771-781. doi: 10.1016/j.surg.2016.09.022.) to these definitions for bleeding in this chart:

• Oozing - Vibe 1 >1-5ml/min

• Mild - Vibe 2 >5-10ml/min

• Moderate - Vibe 2-3

• Severe - Vibe 3 >10-50ml/min (not Vibe 4)

Except for the 1 outlier that was reported to have severe bleeding; All the patients bleeding did fall into the labeling. Probably this severe bleeding patient should have been excluded from the study as this patient is an outlier - although, this patient apparently did not meet the specific criteria to be excluded by the surgeon - according to the blood loss classification of the ATLS classification chart. The patient was not a Class II-IV.

Intraoperative Exclusion Criteria:

Subject has any major intraoperative bleeding incidences (i.e., American College of Surgeons Advanced Trauma Life Support Class II, III or IV Hemorrhage [2013])

ATLS classification of blood loss is classified into four classes based on the estimated percentage of blood volume loss. Class I involves up to 15% blood loss (750ml), Class II involves 15-30% (750-1500ml), Class III involves 30-40% (1500-2000ml), and Class IV involves >40% blood loss (>2000ml). These classifications correlate with changes in vital signs like heart rate and blood pressure. (Spahn, D.R., Bouillon, B., Cerny, V. et al. Management of bleeding and coagulopathy following major trauma: an updated European guideline. Crit Care 17, R76 (2013). https://doi.org/10.1186/cc12685).

2. Achieving hemostasis at 10 minutes was the primary endpoint, but 10 minutes is considered long for the primary endpoint in time to hemostasis studies in the current environment. Supplying a table with the results for the degree of hemostasis at all observed time points would be helpful.

While 10 minutes may not represent the current clinical study bleeding assessment criteria, this study was started in 2015. If we apply the Vibe bleeding scale rates (Lewis KM, Li Q, Jones DS, Corrales JD, Du H, Spiess PE, Lo Menzo E, DeAnda A Jr. Development and validation of an intraoperative bleeding severity scale for use in clinical studies of hemostatic agents. Surgery. 2017 Mar;161(3):771-781. doi: 10.1016/j.surg.2016.09.022.) to the bleeding scale established for this study the majority of patients would bleed 1-10ml/min so over 10 minutes that would be 10-100mls max. None of these patients even meet criteria of Class 1 of 750ml in the STS ATLS Trauma bleeding scale.

While we do not have this data available, we provide in the article the data for time to hemostasis: The time required to achieve hemostasis was lower in the AMP group (1.91±1.15 minutes) than in the SC group (2.28±2.09 minutes), although not significant (p=.309).

3. The bleeding severity score use was only semi-quantitative. Providing a discussion of why a quantitative bleeding severity score was not employed, what such scores exist, and how not using one may affect results would be appropriate.

For this study the investigators decided to devise a study specific bleeding severity score because in 2015 there were no validated quantitative bleeding scales published and peer reviewed. The only validated quantifiable bleeding chart available at the time was the STS ATLS Trauma bleeding scale from 2013.

There were bleeding scales used in many clinical trials. We acknowledge that they are

⦁ “Subjective” description of bleeding

⦁ Valid for investigators only within a particular clinical specialty

⦁ Valid for only a particular type of intervention

For example:

Oz et al used a bleeding scale in cardiovascular surgery in 2000 to describe the initial level of bleeding so as to evaluate the efficacy of a flowable hemostat.

The bleeding severity at each site was characterized as "oozing" or "heavy bleeding" (flowing or spurting). They considered success at 3 minutes and at 10 minutes by severity of bleeding. (Oz MC, Cosgrove DM 3rd, Badduke BR, Hill JD, et al. Controlled clinical trial of a novel hemostatic agent in cardiac surgery. The Fusion Matrix Study Group. Ann Thorac Surg. 2000 May;69(5):1376-82.)

Izzo et al used a bleeding scale for hepatic surgery in 2008 to measure and evaluate the efficacy of a hemostatic matrix before and after treatment.

A hemostatic matrix was applied directly to areas of bleeding. The severity of bleeding before and after application was graded on a 5-point scale (0 = no bleeding, 1 = oozing, 2 = moderate blood flow, 3 = heavy blood flow, 4 = spurting blood). The time to complete hemostasis was also recorded. After application of the gelatin matrix, bleeding severity was assessed at 1-min intervals until hemostasis was complete. (Izzo F, Di Giacomo R, Falco P, Piccirillo M, et al. Efficacy of a hemostatic matrix for the management of bleeding in patients undergoing liver resection: results from 237 cases. Curr Med Res Opin. 2008 Apr;24(4):1011-5.)

The first quantitative validated bleeding scale for intraoperative bleeding is the one published by Lewis KM et al. in 2017 (Lewis KM, Li Q, Jones DS, Corrales JD, Du H, Spiess PE, Lo Menzo E, DeAnda A Jr. Development and validation of an intraoperative bleeding severity scale for use in clinical studies of hemostatic agents. Surgery. 2017 Mar;161(3):771-781. doi: 10.1016/j.surg.2016.09.022.), but as previously mentioned this study started before it was published.

4. Addition to the discussion of how electrocautery was the SC when AMP-SPH is approved for use when conventional treatments are ineffective or impractical. Explaining why it would not have been more appropriate to use another established hemostat (for example a flowable) as the SC in all SC subjects should be discussed.

Electrocautery has long been recognized as a standard method for achieving hemostasis in various surgical procedures. Its widespread use and acceptance in clinical practice make it a relevant control for evaluating new hemostatic agents. Electrocautery is effective for controlling bleeding in many situations, particularly in cases of minor to moderate bleeding, and is often the first-line approach in surgical settings.

The primary aim of our study was to assess the efficacy of AMP-SPH in situations where conventional treatments may be insufficient. While AMP-SPH is approved for use when conventional methods are ineffective or impractical, it is essential to establish its effectiveness against a widely accepted standard. The choice of electrocautery as the control group reflects real-world clinical scenarios where surgeons often rely on this method. By comparing AMP-SPH to electrocautery, we can provide insights into its potential advantages in practice, particularly in complex cases where bleeding control is critical.

While flowable hemostatic agents are indeed established alternatives, they may not be universally applicable in all surgical contexts. The choice of hemostatic agent often depends on the specific surgical procedure, the nature of the bleeding, and the surgeon's preference. Additionally, many flowable hemostatic devices also contain thrombin making the device an “Active” hemostatic device which is in a different category than AMP-SPH which is considered a “Passive” hemostat. Electrocautery remains a versatile tool that can be employed in a variety of situations, making it a suitable control for our study.

We acknowledge the importance of exploring the comparative effectiveness of AMP-SPH against other established hemostatic agents in future studies. This could provide a more comprehensive understanding of its role in the surgical armamentarium and help refine guidelines for hemostatic management in various clinical settings.

5. Reporting of the incidence of rebleeding after 10 minutes intraoperatively.

In the AMP-SPH group, only 1 patient did not achieve hemostasis within 10 minutes. Bleeding was controlled after the 10 min by other methods (not described).

In the SC group, 3 patients did not achieve hemostasis within 10 minutes. Bleeding was controlled in 1 patient with AMP-SPH, in another with sutures, and in the last one by other methods not described by the investigator.

During this study, no other rebleeding events were described intraoperatively and no patients required reintervention for post- operative bleeding.

The following paragraph was added in the results section: “In the AMP-SPH group, hemostasis within 10 minutes was achieved in all but one patient; bleeding in this case was subsequently controlled by unspecified methods. In the SC group, three patients failed to achieve hemostasis within 10 minutes. Bleeding was controlled in these cases using AMP-SPH (n=1), sutures (n=1), and unspecified methods (n=1). No intraoperative rebleeding events or postoperative reinterventions for bleeding were reported during the study”.

6. Explanation of what were the pre-analysis criteria for statistical non-inferiority.

We acknowledge the study limitation of the lack of statistically valid sample size justification. This was added as a study limitation.

The current study was a Phase IV, observational (post-market) study designed to provide descriptive and exploratory data rather than confirmatory evidence. As stated in the study protocol, the sample size was not determined to achieve a specific statistical power to detect differences between the AMP-SPH and Standard of Care (SoC) patient populations. Instead, this study followed an observational, case-controlled design, in which patients were grouped based on treatment exposure rather than through true randomization, which is characteristic of experimental studies such as randomized controlled trials (RCTs). The AMP-SPH group represented the case study cohort, while the SoC group served as the control, allowing for comparative assessment of specific clinical characteristics between the two cohorts.

Given the exploratory nature of this study, our primary objective was to gather preliminary evidence rather than to definitively establish non-inferiority. Although prospective sample size calculation is the preferred methodological approach, as the reviewer correctly noted, a post-hoc analysis was conducted to estimate the number of patients required to detect significant differences based on the observed results. This analysis provided valuable insights into the potential adequacy of the study’s sample size, reinforcing the descriptive and hypothesis-generating role of this investigation.

The following paragraph was added as a study limitation: “Another important limitation of this study is the lack of a statistically valid sample size justification. While a prospective sample size calculation would have been the ideal approach, a post-hoc analysis was performed to estimate the number of patients required to detect significant differences based on the observed results. With an alpha risk of 0.05 in a two-sided test and sample sizes of 44 patients in the AMP-SPH group and 46 in the SC group, the statistical power to detect the observed differences as significant was only 16%. Furthermore, to achieve a power of 80% with an alpha of 0.05, a total of 353 patients per group would have been required for the observed differences in the proportion of patients achieving hemostasis within 10 minutes to reach statistical significance.

7. Clarification as to why all subjects were not able to be included in the final analysis.

The Per Protocol (PP) population includes all subjects who were randomized and

treated with PerClot, and had no major protocol deviations, where major protocol

deviations are defined as:

• Failure to meet any preoperative inclusion/exclusion criteria;

• Failure to meet any intraoperative inclusion/exclusion criteria; or

• Any informed consent violation.

The patients excluded from the PP population had major protocol deviations. Four (4) deviations were determined to be subjects not meeting the Inclusion Criteria or meeting the Exclusion Criteria of the study and the remaining deviations were Informed Consent Form deviations. All subjects were consented to the study, but there were various deviations for the method and/or process of consenting, which included signing the wrong version of the informed consent, the incorrect person dating the consent, or an unauthorized Investigator signing the consent. It was verified that the changes in the different versions of the informed consent were minor and would not have impacted the safety of the patient.

8. Discussion of why there was a trend toward more hematomas in the AMP-SPH group.

In our study, the most commonly reported AE in the AMP group was hematoma, which occurred in 5 cases: 2 cases of pelvic hematoma (one with vaginal spotting); 1 Douglas hematoma; 1 vaginal cuff hematoma; and 1 cupula hematoma. There was one occurrence of hematoma in the SC group. This translated into an incidence of 11.4% in the AMP group and 2.2% in the SC group, but despite this, the difference was not statistically significant.

The following paragraph was added in the discussion section: “The incidence of hematoma formation following pelvic gynecological surgery is highly variable, ranging from

---

## [Decision Letter · Decision Letter 2]

25 Jul 2025

PONE-D-25-02293R2Effectiveness and safety of an absorbable modified polymer starch powder hemostat versus usual care in gynecology procedures: A prospective, multi-center, and randomized study.PLOS ONE

Dear Dr. Garcia-Tejedor,

Thank you for submitting your manuscript to PLOS ONE. After careful consideration, we feel that it has merit but does not fully meet PLOS ONE’s publication criteria as it currently stands. Therefore, we invite you to submit a revised version of the manuscript that addresses the points raised during the review process.

We look forward to receiving your revised manuscript.

Kind regards,

Hideto Sano

Academic Editor

PLOS ONE

Journal Requirements:

Reviewers' comments:

Reviewer's Responses to Questions

**Comments to the Author**

1. If the authors have adequately addressed your comments raised in a previous round of review and you feel that this manuscript is now acceptable for publication, you may indicate that here to bypass the “Comments to the Author” section, enter your conflict of interest statement in the “Confidential to Editor” section, and submit your "Accept" recommendation.

Reviewer #1: All comments have been addressed

Reviewer #2: All comments have been addressed

Reviewer #3: (No Response)

Reviewer #4: All comments have been addressed

2. Is the manuscript technically sound, and do the data support the conclusions?

Reviewer #1: (No Response)

Reviewer #2: Yes

Reviewer #3: Yes

Reviewer #4: Yes

3. Has the statistical analysis been performed appropriately and rigorously? 

Reviewer #1: (No Response)

Reviewer #2: Yes

Reviewer #3: Yes

Reviewer #4: Yes

4. Have the authors made all data underlying the findings in their manuscript fully available?

Reviewer #1: (No Response)

Reviewer #2: Yes

Reviewer #3: Yes

Reviewer #4: Yes

5. Is the manuscript presented in an intelligible fashion and written in standard English?

Reviewer #1: (No Response)

Reviewer #2: Yes

Reviewer #3: Yes

Reviewer #4: Yes

6. Review Comments to the Author

Reviewer #1: (No Response)

Reviewer #2: WELL DESIGNED AND WRITTEN STUDY REGARDING HEMOSTATIC AGENTS WHICH DO HAVE PROFOUND IMPACT IN OUR DAILY PRACTICE

Reviewer #3: In this second revision of the submission, Dr. Garcia-Tejedor and colleagues have contrasted the use of absorbable modified polymer starch powder hemostat (AMP-SPH) compared to standard of care (SC).

The submission has been significantly improved.

An additional modification would further strengthen the submission.

1. Provide the time span over which patient enrollment occurred in terms of chronological dates.

Reviewer #4: The authors of the manuscript have addressed all possible reviewers concerns. Nothing else to comment

7. PLOS authors have the option to publish the peer review history of their article (what does this mean? ). If published, this will include your full peer review and any attached files.

**Do you want your identity to be public for this peer review?** For information about this choice, including consent withdrawal, please see our Privacy Policy .

Reviewer #1: No

Reviewer #2: No

Reviewer #3: No

Reviewer #4: No

---

## [Author Response · Author response to Decision Letter 3]

25 Jul 2025

Response to the reviewers, manuscript PONE-D-25-02293R2, entitled

“Effectiveness and safety of an absorbable modified polymer starch powder hemostat versus usual care in gynecology procedures: A prospective, multi-center, and randomized study”.

Dear Editor-in-Chief, I have considered the comments made by the reviewers and I hope that the paper improves greatly thanks to these comments.

I am sending you the revised manuscript and the rebuttal letter providing a point-by-point response to each of the numbered reviewer comments.

# Review Comments to the Author

Reviewer #1: (No Response)

Reviewer #2: WELL DESIGNED AND WRITTEN STUDY REGARDING HEMOSTATIC AGENTS WHICH DO HAVE PROFOUND IMPACT IN OUR DAILY PRACTICE

Thank you very much indeed for the comment, we highly appreciate it.

Reviewer #3: In this second revision of the submission, Dr. Garcia-Tejedor and colleagues have contrasted the use of absorbable modified polymer starch powder hemostat (AMP-SPH) compared to standard of care (SC).

The submission has been significantly improved.

An additional modification would further strengthen the submission.

1. Provide the time span over which patient enrollment occurred in terms of chronological dates.

The following paragraph was added in the results section: “A total of 90 patients were enrolled across two study sites between November 2015 and July 2017. Screening concluded in July 2017 upon reaching the target enrollment. The study duration was 21 months, from the first patient’s initial visit to the last patient’s final visit. Of the 90 enrolled participants, 44 (48.9%) were assigned to the AMP group and 46 (51.1%) to the SC group (Figure 1).”

Reviewer #4: The authors of the manuscript have addressed all possible reviewers concerns. Nothing else to comment

Thank you very much indeed for the comment, we highly appreciate it.

---

## [Decision Letter · Decision Letter 3]

15 Aug 2025

Effectiveness and safety of an absorbable modified polymer starch powder hemostat versus usual care in gynecology procedures: A prospective, multi-center, and randomized study.

PONE-D-25-02293R3

Dear Dr. Garcia-Tejedor,

We’re pleased to inform you that your manuscript has been judged scientifically suitable for publication and will be formally accepted for publication once it meets all outstanding technical requirements.

Kind regards,

Hideto Sano

Academic Editor

PLOS ONE

Reviewers' comments:

Reviewer's Responses to Questions

**Comments to the Author**

1. If the authors have adequately addressed your comments raised in a previous round of review and you feel that this manuscript is now acceptable for publication, you may indicate that here to bypass the “Comments to the Author” section, enter your conflict of interest statement in the “Confidential to Editor” section, and submit your "Accept" recommendation.

Reviewer #1: All comments have been addressed

Reviewer #3: All comments have been addressed

Reviewer #4: All comments have been addressed

2. Is the manuscript technically sound, and do the data support the conclusions?

Reviewer #1: (No Response)

Reviewer #3: (No Response)

Reviewer #4: Yes

3. Has the statistical analysis been performed appropriately and rigorously? 

Reviewer #1: (No Response)

Reviewer #3: (No Response)

Reviewer #4: Yes

4. Have the authors made all data underlying the findings in their manuscript fully available?

Reviewer #1: (No Response)

Reviewer #3: (No Response)

Reviewer #4: Yes

5. Is the manuscript presented in an intelligible fashion and written in standard English?

Reviewer #1: (No Response)

Reviewer #3: (No Response)

Reviewer #4: Yes

6. Review Comments to the Author

Reviewer #1: (No Response)

Reviewer #3: The authors of the manuscript have addressed all the suggested changes and there are no other comments.

Reviewer #4: THE AUTHORS OF THE MANUSCRIPT HAVE ADDRESSED ALL THE REVIEWERS COMMENTS AND SUGGESTIONS. NOTHING ELSE TO ADD FURTHER

7. PLOS authors have the option to publish the peer review history of their article (what does this mean? ). If published, this will include your full peer review and any attached files.

**Do you want your identity to be public for this peer review?** For information about this choice, including consent withdrawal, please see our Privacy Policy .

Reviewer #1: No

Reviewer #3: No

Reviewer #4: No

---

## [Editor Report · Acceptance letter]

PONE-D-25-02293R3

PLOS ONE

Dear Dr. Garcia-Tejedor,

I'm pleased to inform you that your manuscript has been deemed suitable for publication in PLOS ONE. Congratulations! Your manuscript is now being handed over to our production team.

Kind regards,

on behalf of

Dr. Hideto Sano

Academic Editor

PLOS ONE